A psychrometric model to assess the biological decay of the SARS-CoV-2 virus in aerosols

http://orcid.org/0000-0002-6460-9937 Beggs Clive B. 1 c.beggs@leedsbeckett.ac.uk
http://orcid.org/0000-0003-4411-1493 Avital Eldad J. 2
1 Carnegie School of Sport, Leeds Beckett University , Leeds , UK
2 School of Engineering and Materials Science, Queen Mary University of London , London , UK
Anderson Todd
Electronic publication date: 2021 Mar 2
Publication date: 2021
Volume: 9
Electronic Location ID: e11024
Received 2020 Dec 22; Accepted 2021 Feb 8
Copyright: © 2021 Beggs and Avital
Copyright year: 2021
Copyright holder: Beggs and Avital
License: This is an open access article distributed under the terms of the Creative Commons Attribution License, which permits unrestricted use, distribution, reproduction and adaptation in any medium and for any purpose provided that it is properly attributed. For attribution, the original author(s), title, publication source (PeerJ) and either DOI or URL of the article must be cited.
License URL: https://creativecommons.org/licenses/by/4.0/

Keywords: SARS-CoV-2, Biological decay, Aerosols, Psychrometric model

Funding: UK Royal Academy of Engineering EXP2021/1/247 This work was supported by the UK Royal Academy of Engineering under the programme Engineering X Pandemic Preparedness, grant EXP2021/1/247. The funders had no role in study design, data collection and analysis, decision to publish, or preparation of the manuscript.

==============================
There is increasing evidence that the 2020 COVID-19 pandemic has been influenced by variations in air temperature and humidity. However, the impact that these environmental parameters have on survival of the SARS-CoV-2 virus has not been fully characterised. Therefore, an analytical study was undertaken using published data to develop a psychrometric model to assess the biological decay rate of the virus in aerosols. This revealed that it is possible to describe with reasonable accuracy (R2 = 0.718, p < 0.001) the biological decay constant for the SARS-CoV-2 virus using a regression model with enthalpy, vapour pressure and specific volume as predictors. Applying this to historical meteorological data from London, Paris and Milan over the pandemic period, produced results which indicate that the average half-life of the virus in aerosols outdoors was in the region 13–22 times longer in March 2020, when the outbreak was accelerating, than it was in August 2020 when epidemic in Europe was at its nadir. However, indoors, this variation is likely to be much less. As such, this suggests that changes in virus survivability due the variations in the psychrometric qualities of the air might influence the transmission of SARS-CoV-2.

Introduction

There is increasing evidence that the COVID-19 pandemic may have a seasonal component that is influenced by environmental factors (Audi et al., 2020; Hoque, Saima & Shoshi, 2020; Ma et al., 2020; Merow & Urban, 2020; Moriyama, Hugentobler & Iwasaki, 2020; Qi et al., 2020), similar to that observed for influenza (Chan et al., 2009; Hemmes, Winkler & Kool, 1960; Shaman & Karspeck, 2012; Shaman & Kohn, 2009). For example in Hubei, China, it has been found that a 1 °C increase in the air temperature was associated with a 36–57% decrease in the daily COVID-19 cases when the relative humidity (RH) was in the range 67.0–85.5%. Similarly, a 1% increase in RH led to a 11–22% decrease in the daily confirmed cases when the air temperature was in the range 5.0–8.2 °C. However, these associations were not consistent throughout mainland China (Qi et al., 2020). In Bangladesh, higher air temperatures and higher RH levels have also been significantly associated with a reduction in the transmission of COVID-19 (Haque & Rahman, 2020). Indeed, it has been found that worldwide the SARS-CoV-2 virus tends to spread more in regions that are cooler, with an average temperature of 5–11 °C, and drier, with an absolute humidity (AH) of 4–7 g/m3, suggesting that COVID-19 might exhibit seasonal behaviour (Sajadi et al., 2020). Collectively, these findings suggest that COVID-19 might behave similarly to influenza, which is also caused by an enveloped RNA virus. The influenza A virus is known to survive in aerosols for much longer when AH and vapour pressure (VP) levels are low (Shaman & Kohn, 2009)—something that is possibly due to the ordering of the phospholipid envelope contributing to viral stability, leading to increased transmissibility at lower air temperatures (Polozov et al., 2008).

The reasons for the seasonal variations in the behaviour of COVID-19 are unclear, but may be related to variations in air temperature (Chin et al., 2020; Ma et al., 2020; Qi et al., 2020; Riddell et al., 2020; Wang et al., 2020), humidity (Ma et al., 2020; Qi et al., 2020; Wang et al., 2020) and ultraviolet (UV) radiation from sunlight (Merow & Urban, 2020; Sfica et al., 2020; Tang et al., 2020; Whittemore, 2020), as well as immunological (Moriyama, Hugentobler & Iwasaki, 2020; Whittemore, 2020) and behavioural (Kanzawa et al., 2020) changes. However, because of the many confounding factors that can affect the spread of COVID-19, it is difficult to identify the extent to which each of these environmental factors has influenced the course of the epidemic. For example while it is known that both higher air temperatures (Qi et al., 2020; Wang et al., 2020) and increased UV-B radiation in sunlight (Sfica et al., 2020; Tang et al., 2020; Whittemore, 2020) are associated with reduced transmission of the SARS-CoV-2 virus, because in many parts of the world higher air temperatures are closely associated with increased UV-B radiation it becomes difficult to distinguish between the two. The situation is further complicated by the fact that many researchers misunderstand the concept of RH, which is a ratio of vapour pressures rather than an absolute value. RH is actually the ratio of the observed VP to saturated vapour pressure (SP) expressed as a percentage and as such is strongly affected by temperature because air can hold much more moisture at higher temperatures compared with low temperatures. Unfortunately, researchers do not always appreciate this fact, with the result that it is sometimes reported that during the winter months the air is more humid than in the summer because the RH values are higher, when in fact the air is actually much drier in winter due to SP being considerably lower. Furthermore, many researchers have performed statistical analysis on the RH, failing to appreciate that because it is a ratio involving SP, it is actually a function of air temperature (see Eq. (1) in the methods section below) and therefore not an independent variable. Similarly, because air expands in volume as it warms up, it means that AH, which is the mass of moisture (grams of water) contained in 1 m3 of dry air, is also not independent, but rather, a function of both VP and air temperature. As such, there is a risk that questionable conclusions may have been reached. Consequently, although a clear association exists between COVID-19 prevalence and air temperature and humidity, the precise nature of that association and the reasons for it are much less clear. Given this confusion, we designed the study presented here using published experimental and meteorological data (both in the public domain) with the aim of quantifying the extent to which changes in the psychrometric quality of the air (i.e. changes associated with temperature and humidity) influence the biological survival of the SARS-CoV-2 virus in aerosols. We did this because it is now recognised by the Centers for Disease Control and Prevention (CDC) in the USA (Centers for Disease Control and Prevention, 2020) and the Scientific Advisory Group for Emergencies (SAGE) in the UK (SAGE, 2020) that ‘far-field’ transmission of COVID-19 can occur due the inhalation of small aerosolised respiratory droplets that can remain suspended in the air for considerable periods of time (Beggs, 2020; Miller et al., 2020), especially in poorly ventilated room spaces (RAMP Task 7 Members, 2020; SAGE-EMG, 2020). Therefore, there is a need to better understand how long the SARS-CoV-2 virus can remain viable in aerosols and the extent to which variations in air temperature and humidity influence biological longevity.

Methods

A search of the relevant scientific literature (i.e. published literature, pre-prints and relevant websites) was undertaken to identify published data relating to the survival of the SARS-CoV-2 virus in aerosols under various environmental conditions. Only experiments conducted in the dark were included in the study, with those conducted in the presence of UV light excluded. From each study, the reported biological decay constant, k (min−1), together with the mean air temperature and RH used during experimentation were extracted and compiled into a dataset. With regard to this, the survival of the virus can be computed using the following first order decay equation.

(1) Nt=N0×e(−k.t)

where: N0 and Nt are the number of viable viral particles (virions) at time zero and t minutes respectively; and t is time in minutes.

From the reported mean air temperature and RH values, the psychrometric parameters saturated vapour pressure (SP), vapour pressure (VP), absolute humidity (AH), specific volume (SV), and specific enthalpy, h, were computed using the following empirical equations:

Saturated vapour pressure (Alduchov & Eskridge, 1996): (2) ps=0.61078×e(17.2694×TT+237.29)

where; ps is saturated vapour pressure (kPa) and T is air temperature (°C).

Vapour pressure: (3) pv=ϕr×ps100

where; pv is vapour pressure (kPa) and ϕr is relative humidity (%).

Absolute humidity (Hoque, Saima & Shoshi, 2020): (4) ϕa=2.16679×ϕr×ps100×(273.15+T)

where ϕa is absolute humidity (kg of moisture per m3 of dry air).

Moisture content: (5) g=0.622×pvpb−pv

where; g is moisture content (kg of moisture per kg of dry air) and pb is barometric pressure (i.e. 101.325 kPa).

Specific volume (CIBSE, 2001): (6) s=(0.287+0.461×g)×(273.15+Tpb)

where; s is the specific volume per kilogram of dry air (m3/kg)

Specific enthalpy (Toolbox E, 2020): (7) h=(1.007×T−0.026)+g×(2,501+1.84×T)

where; h is specific enthalpy (kJ/kg).

Having computed the above psychrometric variables, Pearson correlation analysis was performed. The impact of using artificial saliva on the k values was also assessed using a one-way ANOVA. All the statistical analysis was performed using R (R: A language and environment for statistical computing. R Foundation for Statistical Computing, Vienna, Austria), with p < 0.05 deemed as significant.

Multiple linear regression analysis was then performed with the decay constant, k, as the response variable and SP, VP, enthalpy, and SV as predictor variables. RH was not included because it is wholly described by VP and SP and therefore not independent. Similarly, AH was not used because it too can be wholly described by VP (Eq. (5)) and SV (the inverse of density). Instead, VP was used, rather than moisture content, because it is the more fundamental measure. Refinement of the models was performed using backward exclusion, with only variables exhibiting p < 0.1 retained. Heteroscedasticity was evaluated using the Breusch–Pagan test (Breusch & Pagan, 1979), and general applicability assessed using 5-fold cross-validation (CV) (James et al., 2013).

Because three of the experiments in the published literature resulted in negative k values (see Table 1), something that is difficult to interpret, it was decided to build two regression models, one including all the observed k values and another with three negative observations omitted.

Table 1 Reported biological decay constant, k, results for various air conditions, together with computed psychrometric values.

Ref. No.	Artificial Saliva	Decay Constant, k. (min-1)	Mean Temp. (°C)	Mean Relative Humidity (%)	Absolute Humidity (kg/m3)	Vapour Pressure (kPa)	Saturated Vapour Pressure (kPa)	Specific Enthalpy (kJ/Kg)	Specific Volume (kg/m3)	Source	
1	No	0.01060	22.0	65.0	0.0126	1.7185	2.6439	49.402	0.8504	Van Doremalen et al. (2020)	
2	No	0.00618	23.0	53.0	0.0109	1.4890	2.8094	46.728	0.8513	Fears et al. (2020)	
3	Yes	0.01000*	20.0	20.0	0.0035	0.4676	2.3382	27.433	0.8342	Schuit et al. (2020)	
4	Yes	−0.00250*	20.0	37.0	0.0064	0.8651	2.3382	33.708	0.8375	Schuit et al. (2020)	
5	Yes	0.00800*	20.0	53.0	0.0092	1.2393	2.3382	39.659	0.8406	Schuit et al. (2020)	
6	Yes	0.01750*	20.0	70.0	0.0121	1.6368	2.3382	46.031	0.8440	Schuit et al. (2020)	
7	No	0.01500*	20.0	20.0	0.0035	0.4676	2.3382	27.433	0.8342	Schuit et al. (2020)	
8	No	0.01300*	20.0	37.0	0.0064	0.8651	2.3382	33.708	0.8375	Schuit et al. (2020)	
9	No	0.00750*	20.0	53.0	0.0092	1.2393	2.3382	39.659	0.8406	Schuit et al. (2020)	
10	No	0.01500*	20.0	70.0	0.0121	1.6368	2.3382	46.031	0.8440	Schuit et al. (2020)	
11	Yes	−0.01100	10.0	20.0	0.0019	0.2456	1.2279	13.851	0.8040	Dabisch et al. (2020)	
12	Yes	0.01800	10.0	70.0	0.0066	0.8595	1.2279	23.451	0.8089	Dabisch et al. (2020)	
13	Yes	0.00600	20.0	20.0	0.0035	0.4676	2.3382	27.433	0.8342	Dabisch et al. (2020)	
14	Yes	0.01700	20.0	70.0	0.0121	1.6368	2.3382	46.031	0.8440	Dabisch et al. (2020)	
15	Yes	−0.00300	30.0	20.0	0.0061	0.8486	4.2429	43.612	0.8659	Dabisch et al. (2020)	
16	Yes	0.06600	30.0	70.0	0.0212	2.9701	4.2429	78.197	0.8846	Dabisch et al. (2020)	
17	Yes	0.04000	40.0	20.0	0.0102	1.4751	7.3754	63.911	0.9001	Dabisch et al. (2020)	
18	No	0.00910	20.5	55.5	0.0099	1.3384	2.4116	41.755	0.8429	Smither et al. (2020)	
19	Yes	0.01590	20.5	55.0	0.0098	1.3264	2.4116	41.562	0.8428	Smither et al. (2020)	
20	No	0.02270	20.5	86.0	0.0153	2.0740	2.4116	53.614	0.8491	Smither et al. (2020)	
21	Yes	0.04000	20.5	81.0	0.0144	1.9534	2.4116	51.658	0.8481	Smither et al. (2020)	
Note:

* Values estimated from plots.

In order to evaluate the impact of changes in the psychrometric variables on the survival of the SARS-CoV-2 virus in aerosols, the refined (minimum acceptable) regression model exhibiting the lowest Akaike information criterion (AIC) value was used to predict weekly and monthly average k and biological half-life, l0.5, values from historical hourly meteorological data (RP5, 2020) for London, Paris and Milan for the period 1st January–25th October 2020. Where negative values of k were predicted using the hourly data, these were capped at zero (implying no biological decay), before being averaged. The weekly average k values were then compared with published COVID-19 case data acquired for the UK, France and Italy (European Union, 2020) for the period March to October, which approximates to the epidemic period in these countries. As it was only possible to acquire countrywide COVID-19 data, the aim here was simply to compare the relationship between the k values for the cities and the national case data, rather than to draw any firm inference. The monthly average half-life, l0.5, values were computed from the mean monthly k values as follows.

Biological half-life: (8) l0.5=ln(2)k

Results

The results of the literature search are summarised in Table 1, which shows the biological decay constants, k, for the SARS-CoV-2 virus in aerosols, reported for 21 experiments conducted by five research teams (Dabisch et al., 2020; Fears et al., 2020; Schuit et al., 2020; Smither et al., 2020; Van Doremalen et al., 2020). All these researchers used similar methodologies to assess the survival of the virus in aerosols, namely the use of Goldberg type rotating drums into which the SARS-CoV-2 virus was nebulised. Such experiments utilise the competition between gravitational and centrifugal forces induced by the rotating drum to allow aerosol particles to remain airborne for much longer than would be the case in a simple stirred settling chamber (Asgharian & Moss, 1992). The virus was introduced into the drums in aerosols with a diameter range of approximately 2–5 µm and was sampled periodically using impingers (Fears et al., 2020; Smither et al., 2020), or filters (Dabisch et al., 2020; Schuit et al., 2020; Van Doremalen et al., 2020), to determine (following tissue culturing and enumeration) the biological decay rate. For all experiments, the temperature and humidity of the air inside the rotating drums was controlled. Although some researchers (Dabisch et al., 2020; Schuit et al., 2020) also performed experiments in the presence of simulated sunlight, the results of these experiments were excluded from the present study, with only experiments performed in the dark included. Thirteen experiments were conducted with the virus nebulised in artificial saliva (Dabisch et al., 2020; Schuit et al., 2020; Smither et al., 2020), while the rest used a standard tissue culture medium. Two researcher groups (Schuit et al., 2020; Smither et al., 2020) performed experiments using both artificial saliva and a tissue culture medium. However, no statistical difference was found (p = 0.546) between the k values reported for experiments conducted using artificial saliva and those that did not. Table 1 also includes computed psychrometric values for AH, VP, SP, enthalpy, and SV based on the mean air conditions reported for the various experiments.

The correlation results are presented in Table 2. These reveal that the k value was significantly positively correlated with all the psychrometric variables, with the strongest relationships being with AH (r = 0.750; p < 0.001), VP (r = 0.764; p <0.001), enthalpy (r = 0.780; p <0.001) and SV (r = 0.630; p = 0.002). It can also be seen that the correlation between AH and VP was r = 0.999 (p <0.001), indicating that these two variables are very closely related.

Table 2 Correlation r value results.

	k	Temp	RH	AH	Vap. pres	Sat. pres	Enthalpy	Spec. vol	
k	1.000	0.494*	0.436*	0.750***	0.764***	0.489*	0.780***	0.630**	
Temp	0.494*	1.000	−0.182	0.382	0.414	0.962***	0.748***	0.970***	
RH	0.436*	−0.182	1.000	0.794***	0.769***	−0.251	0.476*	0.045	
AH	0.750***	0.382	0.794***	1.000	0.999***	0.290	0.898***	0.593**	
Vap. pres	0.764***	0.414	0.769***	0.999***	1.000	0.325	0.914***	0.621**	
Sat. pres	0.489*	0.962***	−0.251	0.290	0.325	1.000	0.667***	0.915***	
Enthalpy	0.780***	0.748***	0.476*	0.898***	0.914***	0.667***	1.000	0.886***	
Spec. vol	0.630**	0.970***	0.045	0.593**	0.621**	0.915***	0.886***	1.000	
Note:

* Denotes p < 0.05.

** Denotes p < 0.01.

*** Denotes p < 0.001.

k, decay constant; Temp, temperature; RH, relative humidity; AH, absolute humidity; Vap. pres, vapour pressure; Sat. pres, saturated vapour pressure; Spec. vol, specific volume.

The results of the linear regression analysis (utilising the predictor variables temperature, enthalpy, SP, SV, artificial saliva and VP) are presented in Table 3, which shows two competing minimum adequate models (Model 1 and Model 2; see Eqs. (9) and (10) below), one utilising all the data points (n = 21) and the other utilising just the positive k values (n = 18).

Table 3 Refined multiple linear regression models with the biological decay constant, k, as the response variable.

Model	No. of observations (n)	Response Variable	Predictor Variables	Coefficient b (95% CI)	Significance (p value)	Model AIC (p value)	Model R2 (mae)	5-fold CV R2 (mae)	
Model 1	21	k	Intercept	16.9803 [1.8804–32.0802]	0.030	−129.9 (<0.001)	0.718 (0.007)	0.485 (0.009)	
			Enthalpy (h)	0.0622 [0.0075–0.1170]	0.028				
			Vap. pres (VP)	−0.7960 [−1.5103 to −0.0817]	0.031				
			Spec. vol (SV)	−21.9500 [−41.4471 to −2.4529]	0.030				
Model 2	18	k	Intercept	26.2888 [14.7640 to 37.8137]	<0.001	−125.9 (<0.001)	0.877 (0.004)	0.620 (0.019)	
			Enthalpy (h)	0.0923 [0.05110 – 0.13334]	<0.001				
			Vap. pres (VP)	−1.1786 [−1.7141 to −0.6431]	<0.001				
			Sat. pres (SP)	−33.9240 [−48.7952 to −19.0528]	<0.001				
			Spec. vol (SV)	0.0172 [0.0061–0.0282]	0.005				
Note:

k, decay constant; Vap. pres, vapour pressure; Sat. pres, saturated vapour pressure; Spec. vol, specific volume; AIC, Akaike information criterion; mae, mean absolute error; CV, cross validation.

Model 1: (9) k=16.9803+(0.0622h)−(0.7960pv)−(21.9500s)

Model 2: (10) k=26.2888+(0.0923h)−(1.1786pv)−(33.9240pv)+(0.0172s)

These models reveal a strong linear relationship between the decay constant and the predictor variables, irrespective of whether or not the negative k value observations were included. While Model 1 (n = 21) achieved a good fit (R2 = 0.718; mean absolute error (mae) = 0.007), an even stronger fit (R2 = 0.877; mae = 0.004) was observed for Model 2 (n = 18). Nevertheless, because Model 1 exhibited a lower AIC score (AIC = −129.9) than Model 2 (AIC = −125.9), it suggests that Model 1 is likely to exhibit better predictive qualities than Model 2. The Breusch–Pagan test showed no heteroscedasticity problems for either model. CV analysis revealed the cross-validated R2 values to be 0.485 and 0.620 for models 1 and 2 respectively, implying that both have general applicability. Interestingly, when the variables, enthalpy, VP, SP and SV were individually regressed onto the k values using the entire dataset (n = 21), the resulting models were weaker, exhibiting for: enthalpy (R2 = 0.609); VP (R2 = 0.583); SP (R2 = 0.239); and SV (R2 = 0.397).

The results for the two models are graphically presented in Figs. 1 and 2, which show the regression plots for the various studies aggregated together. From these, it can be seen that for both models the observed k values for the various experiments all lie close to the linear best-fit lines, suggesting that the experimental data is consistent and lacking in outliers.

Figure 1 Scatter plot of actual and predicted k values for the various experiments using Model 1 (n = 21).

Figure 2 Scatter plot of actual and predicted k values for the various experiments using Model 2 (n = 18).

The computed mean monthly psychrometric data for the period 1st January to 25th October 2020 for London, Paris and Milan are summarised in Table 4, whereas Table 5 shows the predicted mean monthly k and half-life values produced by applying Model 1, which exhibited the lowest AIC value, to the data in Table 4. From this it can be seen that the predicted biological half-life of the SARS-CoV-2 virus in aerosols during the winter months and early spring is much longer than that for the summer months. For example in Milan during March, when the epidemic first took hold in Italy, the mean biological half-life of the virus was 517.2 min, whereas in August it was only 25.8 min—an approximate 20-fold reduction. A similar pattern was also observed for London and Paris.

Table 4 Average monthly values for the various psychrometric parameters for London, Paris and Milan during the 2020 (January to October).

City	Parameter	January Mean (SD)	February Mean (SD)	March Mean (SD)	April Mean (SD)	May Mean (SD)	June Mean (SD)	July Mean (SD)	August Mean (SD)	September Mean (SD)	October* Mean (SD)	
London	Temp (°C)	7.78	7.99	8.00	12.27	15.09	17.02	18.23	19.93	16.20	11.87	
		(2.80)	(2.89)	(2.95)	(4.52)	(5.25)	(4.37)	(3.67)	(4.80)	(4.19)	(2.47)	
London	RH (%)	83.25	76.46	70.26	62.55	56.26	66.99	66.05	69.56	71.83	83.28	
		(7.92)	(10.84)	(15.35)	(17.13)	(16.67)	(15.98)	(17.73)	(16.83)	(15.89)	(10.60)	
London	Sat. pres (kPa)	1.0734	1.0905	1.0925	1.4854	1.8037	2.0089	2.1454	2.4247	1.9012	1.4067	
		(0.2007)	(0.2111)	(0.2250)	(0.4703)	(0.6253)	(0.6129)	(0.5608)	(0.8051)	(0.5368)	(0.2311)	
London	Vap. pres (kPa)	0.8919	0.8320	0.7559	0.8664	0.9391	1.2776	1.3501	1.5923	1.3118	1.1608	
		(0.1837)	(0.2000)	(0.1986)	(0.1536)	(0.2108)	(0.2478)	(0.2711)	(0.3296)	(0.2841)	(0.1922)	
London	Spec. vol (m3/kg)	0.8028	0.8029	0.8024	0.8154	0.8241	0.8324	0.8365	0.8434	0.8303	0.8167	
		(0.0093)	(0.0096)	(0.0093)	(0.0134)	(0.0159)	(0.0136)	(0.0109)	(0.0153)	(0.0134)	(0.0082)	
London	AH (kg/m3)	0.0069	0.0064	0.0058	0.0066	0.0071	0.0095	0.0100	0.0118	0.0098	0.0088	
		(0.0014)	(0.0015)	(0.0015)	(0.0011)	(0.0015)	(0.0018)	(0.0020)	(0.0024)	(0.0021)	(0.0014)	
London	Enthalpy (kJ/kg)	21.710	20.991	19.805	25.869	29.899	37.239	39.630	45.273	36.953	30.120	
		(5.552)	(5.739)	(5.277)	(5.809)	(7.445)	(7.141)	(5.872)	(8.726)	(7.664)	(5.064)	
Paris	Temp (°C)	6.85	8.93	9.06	15.24	16.68	18.89	20.95	22.43	18.84	12.59	
		(3.59)	(3.08)	(3.54)	(5.08)	(5.09)	(4.89)	(4.30)	(5.46)	(5.06)	(2.71)	
Paris	RH (%)	84.18	76.99	66.70	54.74	55.57	62.65	55.47	60.34	64.52	79.79	
		(9.84)	(10.71)	(19.07)	(18.55)	(16.93)	(17.20)	(16.82)	(18.74)	(17.96)	(11.35)	
Paris	Sat. pres (kPa)	1.0178	1.1646	1.1824	1.8132	1.9872	2.2769	2.5588	2.8534	2.2751	1.4789	
		(0.2433)	(0.2429)	(0.2952)	(0.5900)	(0.6479)	(0.7703)	(0.7405)	(1.0471)	(0.7587)	(0.2701)	
Paris	Vap. pres (kPa)	0.8517	0.8937	0.7809	0.9243	1.0422	1.3250	1.3280	1.5728	1.3669	1.1676	
		(0.2146)	(0.2186)	(0.2666)	(0.2544)	(0.3076)	(0.2405)	(0.2782)	(0.3405)	(0.2839)	(0.2105)	
Paris	Spec. vol (m3/kg)	0.7998	0.8061	0.8056	0.8244	0.8295	0.8382	0.8441	0.8504	0.8384	0.8188	
		(0.0117)	(0.0103)	(0.0115)	(0.0153)	(0.0159)	(0.0148)	(0.0126)	(0.0165)	(0.0156)	(0.0090)	
Paris	AH (kg/m3)	0.0066	0.0069	0.0060	0.0069	0.0078	0.0098	0.0098	0.0115	0.0101	0.0088	
		(0.0016)	(0.0016)	(0.0020)	(0.0019)	(0.0022)	(0.0017)	(0.0021)	(0.0025)	(0.0020)	(0.0015)	
Paris	Enthalpy (kJ/kg)	20.139	22.915	21.276	29.815	33.161	39.906	42.058	47.516	40.536	30.970	
		(6.766)	(6.223)	(6.908)	(7.457)	(8.579)	(7.230)	(6.325)	(8.569)	(8.112)	(5.570)	
Milan	Temp (°C)	3.05	7.04	8.52	13.44	18.13	20.41	23.92	24.47	19.52	12.43	
		(4.54)	(5.31)	(4.96)	(5.88)	(4.21)	(4.87)	(4.15)	(4.51)	(5.25)	(3.87)	
Milan	RH (%)	83.51	66.13	68.84	60.61	66.81	70.89	65.55	64.43	72.32	81.10	
		(19.18)	(26.63)	(23.47)	(24.40)	(20.94)	(19.83)	(17.35)	(18.78)	(19.07)	(17.53)	
Milan	Sat. pres (kPa)	0.7967	1.0640	1.1679	1.6397	2.1448	2.4953	3.0501	3.1672	2.3745	1.4835	
		(0.2689)	(0.3887)	(0.4049)	(0.5979)	(0.5579)	(0.7695)	(0.7659)	(0.8628)	(0.7617)	(0.3756)	
Milan	Vap. pres (kPa)	0.6231	0.6431	0.7469	0.8913	1.3426	1.6358	1.8981	1.9108	1.6198	1.1614	
		(0.1267)	(0.2524)	(0.2373)	(0.3237)	(0.2911)	(0.2654)	(0.3204)	(0.3297)	(0.3693)	(0.2627)	
Milan	Spec. vol (m3/kg)	0.7872	0.7987	0.8037	0.8190	0.8361	0.8452	0.8575	0.8592	0.8425	0.8183	
		(0.0135)	(0.0158)	(0.0149)	(0.0174)	(0.0124)	(0.0146)	(0.0129)	(0.0135)	(0.0169)	(0.0122)	
Milan	AH (kg/m3)	0.0049	0.0050	0.0057	0.0067	0.0100	0.0121	0.0138	0.0139	0.0120	0.0088	
		(0.0010)	(0.0019)	(0.0018)	(0.0024)	(0.0022)	(0.0019)	(0.0023)	(0.0024)	(0.0027)	(0.0019)	
Milan	Enthalpy (kJ/kg)	12.701	17.062	20.187	27.474	39.416	46.451	54.299	55.071	45.297	30.708	
		(5.862)	(7.409)	(7.157)	(8.456)	(6.463)	(7.168)	(7.479)	(7.356)	(9.823)	(6.872)	
Notes:

* Data for 1 to 25th October 2020 only.

Temp, temperature; RH, relative humidity; AH, absolute humidity; Vap. pres, vapour pressure; Sat. pres, saturated vapour pressure; Spec. vol, specific volume; SD, standard deviation.

Table 5 Predicted monthly mean biological decay constant, k, and half-life values for London, Paris and Milan during the 2020.

City	Parameter	January Mean (SD)	February Mean (SD)	March Mean (SD)	April Mean (SD)	May Mean (SD)	June Mean (SD)	July Mean (SD)	August Mean (SD)	September Mean (SD)	October* Mean (SD)	
London	k.pred (min−1)	0.00113	0.00108	0.00078	0.00262	0.00484	0.00963	0.01150	0.01697	0.00980	0.00488	
		(0.00192)	(0.00183)	(0.00158)	(0.00265)	(0.00388)	(0.00528)	(0.00486)	(0.00828)	(0.00607)	(0.00355)	
London	Mean half-life (min)	613.4	641.8	888.6	264.5	143.2	72.0	60.3	40.8	70.7	142.0	
Paris	k.pred (min−1)	0.00110	0.00171	0.00140	0.00500	0.00704	0.01143	0.01287	0.018.23	0.01224	0.00545	
		(0.00192)	(0.00254)	(0.00229)	(0.00359)	(0.00548)	(0.00562)	(0.00530)	(0.00830)	(0.00639)	(0.00380)	
Paris	Mean half-life (min)	630.1	405.3	495.1	138.6	98.5	60.6	53.9	38.0	56.6	127.2	
Milan	k.pred (min−1)	0.00008	0.00077	0.00134	0.00442	0.01142	0.01756	0.02626	0.02686	0.01762	0.00565	
		(0.00035)	(0.00149)	(0.00208)	(0.00386)	(0.00521)	(0.00722)	(0.00870)	(0.00824)	(0.00800)	(0.00417)	
Milan	Mean half-life (min)	8663.8	900.1	517.2	156.8	60.7	39.5	26.4	25.8	39.3	122.7	
Note:

* Data for 1 to 25th October 2020 only.

When the mean weekly k values for London, Paris and Milan are compared with the COVID-19 case data for the UK, France and Italy (Fig. 3), it can be seen that there is a broadly inverse relationship between the two plots (March to October, r = −0.258 (London); −0.124 (Paris); −0.512 (Milan)), with infections lowest during the summer months when the biological decay constant, k, was at its greatest. However, this relationship only reached significance in the case of Milan and Italy (p = 0.002).

Figure 3 Plots of: (A) the biological decay constant, k, for London, Paris and Milan during the period 1st January to 25th October 2020; (B) weekly COVID-19 cases for the UK, France and Italy during the same period.

Discussion

The principal finding of our study is that the biological decay constant, k, for the SAR-CoV-2 virus in aerosols broadly conforms to a linear relationship and can be modelled with reasonable accuracy using a regression model with the variables: enthalpy, SV and VP (and possibly SP) as predictors. As such, this supports the growing body of evidence that the COVID-19 epidemic is influenced by changes in temperature and humidity (Chin et al., 2020; Ma et al., 2020; Morris et al., 2020; Qi et al., 2020; Riddell et al., 2020; Wang et al., 2020) and suggests that in this respect the SARS-CoV-2 virus behaves similarly to influenza A (Haque & Rahman, 2020; Shaman & Kohn, 2009), which is also an enveloped RNA virus. Indeed, Shaman & Kohn (2009) and Marr et al. (2019) (re-analysing Harper (1961) original data) both produced similar results to ours, demonstrating that the survival of the influenza virus in aerosols is inversely correlated with VP and air temperature. This mirrors our finding that the k value is most strongly correlated (r = 0.780, P < 0.001) with specific enthalpy, h, which is a composite measure representing total heat energy in the air and as such reflects both VP and air temperature. In practical terms, this implies that survival of the SARS-CoV-2 virus in aerosols is strongly influenced by the total energy in the air (i.e. the sensible heat energy of the dry air and the latent heat energy in the evaporated moisture combined). Consequently, as air temperature, VP and enthalpy increase, so the value of k also increases, with the result that the biological half-life, l0.5, decreases and the virus survival time becomes shorter.

The reasons why survival of the SARS-CoV-2 virus should reduce as the air temperature and VP increase are poorly understood. However, the fact that other enveloped viruses such as influenza, respiratory syncytial virus (RSV) and human coronavirus (HCoV), all exhibit seasonal cycles similar to COVID-19 (Moriyama, Hugentobler & Iwasaki, 2020), suggests that structural changes in the phospholipid envelope and surface proteins due to variations in temperature and humidity may be responsible (Marr et al., 2019; Shaman & Kohn, 2009). It may be that low-temperature conditions promote the ordering of lipids in the viral envelope and that this contributes to viral stability (Polozov et al., 2008). However, this does not explain how changes in VP can affect viral stability in respiratory droplets when the actual virions are not exposed to the ambient air and therefore not directly interacting with the water vapour (Marr et al., 2019). It therefore appears likely that evaporation from the droplet surface plays a key role in determining survivability. This is because evaporation affects the chemistry of droplets, which in turn might affect the stability of any viral particles contained within (Lin, Schulte & Marr, 2020; Marr et al., 2019).

The evaporation process is dominated by the difference between the SP at the droplet surface and the ambient air VP (Mittal, Ni & Seo, 2020). It also depends on the temperature difference between the droplet surface and the ambient air; with flow convective effects influenced by the Reynolds, Schmidt and Prandtl numbers playing an important role (Kincaid & Longley, 1989). As such, the droplet evaporation process exhibits a complex dependency on both temperature and humidity, which is difficult to model. However, Shaman & Kohn (2009) developed a simplistic evaporation model which related time and the rate of decrease in the droplet’s radius, dr/dt, with the ratio of the air vapour pressure deficit to the ambient air temperature, (pS − pV)/(273.15 + T), where T is in the range –20 °C to 40 °C. As such, they were able to show that the rate at which volumetric change occurs in aerosol droplets, and by inference changes in the solute (i.e. salts, proteins, etc.) concentration, are largely influenced by VP and air temperature. In particular, changes in pH within the aerosol droplet that are induced by evaporation may trigger conformational changes of the surface glycoproteins in enveloped viruses and subsequently compromise their infectivity (Yang & Marr, 2012). Changes in pH due to droplet evaporation have been shown to be critical importance to the survival of the enveloped bacteriophage φ6 (Lin, Schulte & Marr, 2020).

As well as having a biological effect, evaporation profoundly influences droplet size and thus the aerodynamic behaviour of any respiratory droplets exhaled. With respect to this, respiratory droplets can be classified as being either large (>100–125 µm in diameter) or small (<100–125 µm in diameter). With large droplets the sedimentation process dominates over the evaporation process, with the result that they tend to travel only short distances from the source due to their ballistic behaviour (Mittal, Ni & Seo, 2020; Seminara et al., 2020; Walker et al., 2020). By comparison, small droplets are dominated by evaporation in their initial stage, rapidly reducing in size to become semi-solid aerosol particles containing proteins and salts (Marr et al., 2019; Walker et al., 2020) or fully dry particles called droplet nuclei, both of which can be transported longer distances on convection currents and widely distributed in room spaces (Miller et al., 2020). While the former are largely unaffected by air humidity, the latter are profoundly affected by VP if the flow convective effects are small (Seminara et al., 2020). So as well as affecting virus survival in aerosol droplets, air temperature and humidity can also affect the aerodynamic behaviour of respiratory aerosols and thus the transmission of viral diseases (Shaman & Kohn, 2009). Indeed, Shaman & Kohn (2009) were able to show that increased evaporation at lower VPs produced smaller aerosol particles (i.e. the formation of greater numbers of droplet nuclei) that stay airborne for longer. However, they found no strong correlation between the ratio of VP deficit to temperature and the actual transmission of influenza, suggesting that increased production of airborne droplet nuclei in low-VP conditions may not be the principle means by which VP modulates influenza transmission (Shaman & Kohn, 2009). Rather, they found a much stronger statistically significant relationship between VP and virus survival, suggesting that the modulation of viral survivability in aerosols is the primary means through which VP affects airborne influenza transmission (Shaman & Kohn, 2009). This opinion is however challenged by early studies on influenza in humans, which found that illness could be induced with substantially lower virus titres when administered as a small droplet aerosol rather than by nasal droplets (Alford et al., 1966). This suggests that influenza infection may be induced more efficiently when the virus is deposited in the lower respiratory tract rather than in the upper airways. If this is the case, then small virus laden droplet nuclei might pose a greater threat than larger droplets, or at least contribute to the airborne spread of influenza (Bridges, Kuehnert & Hall, 2003; Yan et al., 2018). Interestingly, several studies have found viral RNA to be more concentrated in small droplet nuclei compared with larger droplets (Bischoff et al., 2013; Coleman & Sigler, 2020; Lindsley et al., 2010; Yan et al., 2018). It may therefore be that the warm dry air conditions that predominate indoors during the winter months are more conducive to the production of small virus laden droplet nuclei and that this promotes the airborne transmission of viral infections within buildings during this period.

While external air conditions may vary greatly throughout the year, it is general practice to maintain air temperatures inside buildings at an approximately constant level with minimal variation, although during the summer, internal air temperatures may rise if air conditioning is not employed. However, while it is possible through the use of heating to maintain comfortable room air temperatures during the winter months, internal RH levels may drop to <30% when external air temperatures are low (e.g. <4 °C) and the air is dry (e.g. VP < 0.75 kPa), something which can have a dramatic impact on the survival of the SARS-CoV-2 virus. This is clearly illustrated in Fig. 4, which shows the predicted (using Model 1) k values for the virus over a range of RH levels for a constant air temperature of 21 °C. From this it can be seen that at 30% RH, k = 0.0063 min−1 (half-life = 112.4 min), whereas at 50% RH the value of k increases to 0.0111 min−1 (half-life = 62.6 min) and at 70% RH, k = 0.0198 min−1 (half-life = 35.0 min). Similarly, if the RH is kept constant, say at 50% (Fig. 5), we can see that varying the air temperature has an even more dramatic impact on the k value. At 10 °C and 50% RH, the model predicts that k = −0.0003 min−1 (i.e. no biological decay), whereas at 25 °C and 50% RH, k = 0.0168 min−1 (half-life = 36.1 min). From this it can be seen that if a room space is maintained at a temperature => 21 °C with an RH => 50%, then this should cause the virus to biologically decay more rapidly than would be the case if the air were cooler and drier.

Figure 4 Predicted k values (using Model 1) for air at 21 °C for a range of humidity levels (RH = 20–70%).

Figure 5 Predicted k values (using Model 1) for air at 50% RH for a range of temperatures (temperature = 10–30 °C).

It is noticeable that for three of the experiments studied, negative k values were reported (see Table 1). This implies that during these experiments the viral load marginally increased rather than decreased, something that is difficult to explain, given that virion numbers in aerosols should in theory only decrease with time. We therefore constructed two competing models: Model 1 using all the data points, and Model 2 utilising only the data for which k was positive. Both demonstrated a strong linear relationship between k and the various predictors. However, although Model 2 produced a better fit than Model 1, we nonetheless decided to use Model 1 to predict k values from the meteorological data because this model exhibited a lower AIC value than Model 2, implying that it would be the better predictive tool. However, Model 1 had the limitation that it yielded negative k values at the extremes when the air was cool and dry, as is the case at 10 °C and 50% RH (see above). Although in the present study, we got around this problem by capping all the predicted negative k values at zero (a reasonable assumption implying no biological decay), it nevertheless implies that the predictive power of the current linear model may be limited at the extremes, particularly at low temperature and RH levels. As such, this mirrors the findings of Yang, Elankumaran & Marr (2012) for influenza and suggests that a more complex two-phase model may be required to accurately predict the value of k when air temperatures and RH levels are low. Notwithstanding this, for most practical purposes, when assessing the survival of the SARS-CoV-2 virus indoors, Model 1 appears to yield a reasonable approximation of k, because air temperatures rarely fall below about 12–15 °C inside most buildings, even when they are intermittently heated (Kane, Firth & Lomas, 2015).

In order to evaluate how changes in weather during the 2020 COVID-19 pandemic might have affected the SARS-CoV-2 biological decay constant, k, and thus survival half-life we applied our regression model (Model 1) to historical meteorological data for London, Paris and Milan (Table 4). This revealed (Table 5) that for all three locations the virus survived in the air for much longer during the winter, autumn and early spring compared with the summer. For example in Milan during March 2020 (when the Italian COVID-19 epidemic started to accelerate) the predicted mean half-life of the virus was 517 min, whereas in July and August (when the Italian epidemic reached its low point) the mean half-life was just 26 min. Mirroring the findings of Morris et al. (2020) and Shaman & Kohn (2009), this suggests that changes in VP and air temperature may contribute to the seasonal fluctuations that have been observed in the COVID-19 epidemic, particularly in temperate regions. Indeed, when the predicted weekly mean k values for these cities are compared with the weekly number of COVID-19 cases for the UK, France and Italy respectively (Fig. 3), it can be observed that a broadly inverse relationship exists in all three countries, with COVID-19 cases lowest when the k values are highest. Of course, any comparison between weather data for individual cities and COVID-19 cases for whole countries can only be of limited value and must therefore be treated with great caution. Nonetheless, despite the fact that transmission of the virus is thought to primarily occur indoors, it is clear that COVID-19 case numbers were lowest when survival of the virus in the outdoor environment was at its shortest. This suggests that external weather conditions affect virus survival in aerosols within buildings, which is to be expected given that room temperature and humidity levels are known to respond to changes in the psychrometric condition of the outside air (Jones, 2007). In particular, because most buildings lack any form of humidification, it is generally the case that the humidity levels experienced inside reflect the vapour pressure levels in the outside air. So, in winter when, for example the outside air condition might be 0 °C and 90% RH (VP = 0.5497 kPa), the RH will be about 23.5% in a room heated to 20 °C, assuming that no additional humidification of the air occurs within the room space. However, in autumn when the outside air might be at, say, 13 °C and 75% RH (VP = 1.1233 kPa), for the same room at 20 °C the RH would be about 48%.

While the results in Table 5 represent the predicted k and half-life values for external air conditions, it should be remembered that COVID-19 is a disease that is primarily transmitted indoors. With this in mind, because indoor air is generally heated during the cooler months, the range of the k values exhibited inside buildings is likely to be considerably less than that experienced outdoors. So for example for buildings located in a temperate region such as the UK, the internal air conditions might range from, say, 18 °C and 25% RH, equating to k = 0.0038 min−1 (half-life = 181.5 min) at one extreme (i.e. a poorly heated space on a cold day in winter), to, say, 25 °C and 60% RH, equating to k = 0.0262 min−1 (half-life = 26.5 min) at the other (i.e. a room space on a hot humid day in summer). From this it is reasonable to suppose that for temperate regions, the half-life of the SARS-CoV-2 virus within buildings could be as much as seven times longer during the winter months compared with the summer, depending, of course, on the weather conditions experienced, the degree to which the indoor space is heated, and the internal moisture gains. Having said this, it should be noted that this represents the extremes of the range and for much of the time the variation in the virus half-life should be less than this.

While in this study we have been able to characterise the relationship that exists between the psychrometric qualities of the air and survival of the SARS-CoV-2 virus in aerosols, it is important to note that we cannot say to what extent this contributes to the overall transmission of the SARS-CoV-2 virus. This is because the seasonal variations that result in large changes in air temperature and VP also coincide with changes in: UV-B irradiation levels; population behaviour; building occupancy levels; and ventilation rates, as well as changes in the human immune system. For example in temperate regions during the summer months, populations spend more time outdoors, as well as ventilating buildings to a greater extent (RAMP Task 7 Members, 2020), both factors that tend to inhibit COVID-19. Likewise, viral degradation due to UV-B radiation in sunlight is greatly increased during the summer months (Ratnesar-Shumate et al., 2020; Schuit et al., 2020; Tang et al., 2020), as are vitamin D levels due to exposure to sunlight (Tang et al., 2020; Whittemore, 2020). Furthermore, the effect of low air temperatures and VP levels during winter on the respiratory tract should not be ignored. Dehydration due to inhaling dry cold air can cause the mucosal layer in the respiratory tract to become more viscous, immobilising the cilia, and reducing the body’s ability to clear pathogens from the airways during wintertime (Moriyama, Hugentobler & Iwasaki, 2020). Notwithstanding this, our finding that the SARS-CoV-2 virus is likely to survive in aerosols for much longer during winter and early spring compared with summer supports the observations of many other researchers (Chin et al., 2020; Morris et al., 2020; Riddell et al., 2020; Schuit et al., 2020; Van Doremalen et al., 2020; Wang et al., 2020). Importantly however, it should be noted that the half-life survival times of >100 min predicted by our model for indoor spaces during the colder months, are much longer than the period virions are likely to remain airborne in a typical room space. As such, this raises two intriguing questions:Does the psychrometric quality of the air alter the viral load in aerosol droplets that are inhaled and does this influence the progression of the epidemic? And;

Does the psychrometric quality of the air alter the survival time and viral load in respiratory droplets that impact on surfaces and hands, etc. and does this in any way influence the transmission of the SARS-CoV-2 virus?

While it would appear reasonable to assume that increased viral load is associated with an increase in the biological half-life, the answers to both these questions are at this stage unknown, and further investigation will be required in order to determine whether or not changes in air temperature and VP significantly affect the transmission of the SARS-CoV-2 virus by either route.

Although we have characterised a strong relationship between survival of the SARS-CoV-2 virus in aerosols and air enthalpy, VP and SV, we are aware that our study has several noticeable limitations. Chief amongst these is our reliance on a limited dataset collected by disparate researchers, using a variety of experimental procedures. Furthermore, when calculating the psychrometric parameters we used the mean temperature and RH values reported in the respective publications, which meant that we could not compute any margins of error. Notwithstanding this, because of the ensemble approach taken and the strengths of the linear relationships observed in Figs. 1 and 2, and in Table 3, which we cross-validated, it suggests that our findings and conclusions are valid. Having said this, we recommend that our results should be considered as being indicative only and that further experimental work should be undertaken to refine our findings. In addition, because our model yielded negative k values at the extremes, which in theory should not occur, it is recommended that further work be undertaken to better understand how the virus behaves in aerosols in cold dry air, so that it can be modelled more accurately under these conditions.

Because our psychrometric model has the potential to yield useful information regarding the survival of the SARS-CoV-2 virus both inside and outside buildings, it is recommended that the analysis presented here be extended in the future to cover typical indoor environments as well as other geographical locations. Also, the analysis should be extended to cover a longer time period than that reported here, as this might yield useful insights into the impact of variations in air temperature and humidity on the progression of the pandemic.

Conclusions

We have been able to demonstrate that survival of the SARS-CoV-2 virus in aerosols is inversely related to both air temperature and VP, with survival greatly increasing during the winter months when the air is cooler and drier. More specifically, we have been able to build a linear model that describes with reasonable accuracy (R2 = 0.718) the biological decay constant, k, of SARS-CoV-2 using enthalpy, VP and SV as predictors. When applied to historical meteorological data for the 2020 COVID-19 pandemic for London, Paris and Milan (Table 5), the results suggest that the average half-life of the virus in aerosols outdoors was in the region 13–22 times longer in March, when the outbreak was accelerating, than it was in August when it was at its nadir. However indoors, although a similar pattern will exist, because room spaces are generally heated, the variation in the virus half-life between summer and winter is likely to be much less than that exhibited outdoors. Nonetheless, the considerable range in virus half-life exhibited throughout the year, suggests that variations in the psychrometric qualities of the air might play an important contributory role in influencing the transmission of the SARS-CoV-2 virus.

Additional Information and Declarations

Competing Interests

Author Contributions

Data Availability

The authors declare that they have no competing interests.

Clive B. Beggs conceived and designed the experiments, performed the experiments, analysed the data, prepared figures and/or tables, authored or reviewed drafts of the paper, and approved the final draft.

Eldad J. Avital performed the experiments, analysed the data, authored or reviewed drafts of the paper, and approved the final draft.

The following information was supplied regarding data availability:

The raw data are available in Tables 1–5.

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
