# Peer review of "A psychrometric model to assess the biological decay of the SARS-CoV-2 virus in aerosols"

_PeerJ, doi:10.7717/peerj.11024_

## Round 0.1 · original submission · Minor Revisions

Please address or respond to the reviewers' comments. In particular, please provide some additional details on the experiments upon which the statistical analysis is based.

·

Basic reporting

Sometimes the disease COVID-19 is used instead of the virus SARS-CoV-2, eg line 47 where the disease is compared with the influenza virus. The authors should check their manuscript and ensure that they use SARS-CoV-2 when referring to the virus, and COVID-19 when referring to the disease. Note, that it is the virus that transmits, not the disease.

Experimental design

no comment

Validity of the findings

No Comment

Additional comments

Line 42 – worth noting that Qi et al did not see similar relationships in other areas of mainland China
Sometimes the disease COVID-19 is used instead of the virus SARS-CoV-2, eg line 47 where the disease is compared with the influenza virus. The authors should check their manuscript and ensure that they use SARS-CoV-2 when referring to the virus, and COVID-19 when referring to the disease. Note, that it is the virus that transmits, not the disease.

Line 89, the authors might also reference the CDC in the US and PHE as two public bodies that have specifically recognised aerosol role in far field transmission of SARS-CoV-2, although to date, (with the exception of aerosol generating procedures) there is not an explicit recognition from WHO.
Line 90, consider referencing SAGE EMG paper too https://www.gov.uk/government/publications/emg-role-of-ventilation-in-controlling-sars-cov-2-transmission-30-september-2020
Line 157 – reference for the Breusch-Pagan test please
Line 241 – the authors should note that this linear relationship is shown for data within air temperature range of 10-40°C and VP of 0.2456 and 2.97 kPa. From the analysed data set we can not be confident that we can extrapolate this relationship outside these values. For example in Yang et al https://doi.org/10.1371/journal.pone.0046789 fig 1 the influenza viability is linear over c RH 50% to 100% but behaves differently at lower RH – obviously this needs translating into vapour pressure, but it is important to note that, in the absence of data, we don’t know if the relationship proposed by the authors remains for data outside the range of data used to generate the relationship.
On further reading I note that the authors do go some way to acknowledge this in lines 377 – 403, but an explicit note on the limits of this relationship would be useful earlier in the discussion around Line 241

Line 287 – a special character has not translated into the pdf correctly
Line 291 and 292 - a special character has not translated into the pdf correctly
Line 324 to 341, although the predicted change in viability of the virus in different climatic conditions is interesting, the authors should note that most transmission is considered to occur indoors – nevertheless, for many indoor ventilation strategies the outside weather and vapour pressure conditions will affect the indoor conditions too. It would be helpful for the authors to note this.

A figure of chosen model showing predicted k changes with temp at fixed RH, and with RH at fixed temp would be a useful addition to this paper

·

Basic reporting

This reviewer found the paper interesting and enlightening, it was well written, with the English professional and simple enough to follow throughout.

The formatting of long reference lists within sentences made for difficult reading within some parts of the document. This I noted was not the case in the pre-print, which made use of numbered references. The following sentence is an example of difficult to read text due to multiple references interrupting the flow of the text. I am not entirely sure if this can be sorted by the authors or the journal?

“The reasons for the seasonal variations in the behaviour of COVID-19 are unclear, but may be related to variations in air temperature (Chin et al. 2020; Ma et al. 2020; Qi et al. 2020; Riddell et al. 2020; Wang et al. 2020), humidity (Ma et al. 2020; Qi et al. 2020; Wang et al. 2020) and ultraviolet (UV) radiation from sunlight (Merow & Urban 2020; Sfica et al. 2020; Tang et al. 2020; Whittemore 2020), as well as immunological (Moriyama et al. 2020; Whittemore 2020) and behavioural (Kanzawa et al. 2020) changes.”

However, the references provided were sufficient, of high quality, up to date and with enough context to enable to reader to review the subject in more detail, as required.

I personally found it a jump from the formula to Figures 1 and 2, and considered whether seeing a tables for these data might improve clarity, or see further explanation from the authors, for not including these data.

No further comments.

Experimental design

The paper has probably filled a knowledge gap and reviewed an area of aerosol science / microbiology, that has received perhaps just a cursory glance beforehand, certainly for COVID-19.

It is an area of I have briefly reviewed and now upon reading this paper I realise I fundamentally misunderstood some of the effects of energy, humidity and temperature on (enveloped) viruses, as did those perhaps writing the articles I read.

As the authors themselves say, it requires someone now to do some research to fill experimental gaps. However, I believe the methodology and investigation has been thorough and seems to fit the datasets available. Brief working through the data suggest a user could reproduce or further develop the models, k and input temperatures & humidity data and come up with half-life survival times for another data-set.

No further comments.

Validity of the findings

Overall, the paper is well rounded and does indeed indicate the SARS-CoV-2 virus is similar to influenza A, another enveloped RNA virus, with seasonal behaviour, and is influenced by variations in air temperature and humidity.

The authors give explanation to the viral characteristics and behaviour that would be effected by lower temperatures and humidity, with an elongated and well referenced discussion.

I was pleased the authors clearly explained the limitation of the work and would hope to see them or others take up the mantle with future work, some thoughts on this include:

- Extending the work beyond October for the Milan, Paris & London.
- Extending the work beyond the above cities.
- Perhaps the authors could consider typical half-lives of the virus within typical indoor scenarios (a typical office, home, etc.), if this is plausible.

No further comments.

Additional comments

Insightful and interesting paper, although I would be keen to see the paper published sooner rather than later, the extensions I have discussed would enhance it still further. Perhaps these extensions could be published as a follow up piece of work, rather than delay the publication of this paper.

No further comments.

·

Basic reporting

Well written paper, generally easy to follow and relevant information provided. However, I suggest that more information would be useful on the experiments upon which the statistical analysis is based. What were the experimental setups, typically? How similar are these experiments to each other? Do they all follow the same/similar procedure? If not are there differences that may affect the decay constant? Maybe the size range of aerosol droplets generated, for example?

While I am able to understand the meaning of the different statistical analyses used in the paper, I can’t say that I’m that familiar with many of these methods, e.g. Breusch-Pagan test. I’m sure readers who are familiar with statistical methods will be familiar with these, however for those who aren’t, like myself, suitable references would be useful when discussing such methods.

Experimental design

The experimental design seems reasonable to me. However my knowledge of statistics is not particularly in depth so I can’t say I’m well placed to critique some of the statistical methods used.

Validity of the findings

The main finding of the paper is that there seems to be a definite, predictable relationship between the psychometric data considered (mainly temperature and humidity) and the value of the decay constant. The authors provide a convincing case regarding this result.
The authors then attempt to apply the derived relationship to historical meteorological data, providing estimates of the virus survival time outdoors. However, airborne transmission is likely to occur mostly indoors. Could the authors comment on the expected correlation between outdoor conditions and indoor conditions, and subsequently the decay constant? For example, indoor temperatures in winter will be significantly higher than outdoors. AH is likely to be more closely correlated between indoors and outdoors. Given the strong correlation between k and AH, I would expect the trend in fig 4 to look similar for indoor conditions? However the half-life periods may be considerably shorter in winter when considering the indoor environment? Is there a simple, indicative way that indoor conditions could be modelled for different seasons?

Given that airborne transmission outdoors is likely to be very low for all seasons, how relevant is the outdoor half life of the virus? Which is what the model is used to predict in this case. With this in mind, I do question the meaning of the comparison between case rates and predicted outdoor decay constant. The authors do acknowledge the limitations of this comparison and some of the other parameters which are likely to be significant such as seasonal variations in sunlight and indoor occupancy. They also state, referring to the decay constant, “it is important to note that we cannot say to what extent this contributes to the overall transmission of COVID-19”. In which case, why try to link case rates and the decay constant? The comparison is further complicated by the gradual increase in testing over the period of the pandemic.

Additional comments

The paper shows a clear link between the psychometric parameters and the decay constant of the virus, which is an important contribution to the literature. It also provides a very interesting and insightful discussion of the possible mechanisms by which these parameters affect the viability of the virus. The authors use the derived model to predict the decay constant using meteorological data since the start of the pandemic. This provides an interesting illustration of the degree with which the decay constant can vary under different conditions, with an order of magnitude difference between seasons. A comparison is made between the number of cases reported in the UK, France and Italy and the predicted decay constants, roughly showing an inverse relation. This comparison is at this point tenuous due to many other factors which are yet to be understood. However the authors do acknowledge this. Overall I think this is a good paper, however is there a more meaningful analysis to be made to illustrate the significance of the derived model?

Eq 8. 0.6931 comes from ln(2). It would clarify the meaning of this equation to write ln(2).

L145 What was the alternative to artificial saliva? It would be useful to comment on the significance of the solution used.

L191. Why control for artificial saliva if no statistical difference was seen between experiments which did and didn’t use it?
L280. should be lower case p?

---

## Round 0.2 · accepted · Accept

Thank you for addressing reviewer comments and for revising your manuscript as appropriate.